# Overview of the *Justicia* Genus: Insights into Its Chemical Diversity and Biological Potential

**DOI:** 10.3390/molecules28031190

**Published:** 2023-01-25

**Authors:** Marcos Rodrigo Beltrão Carneiro, Lóide Oliveira Sallum, José Luís Rodrigues Martins, Josana de Castro Peixoto, Hamilton Barbosa Napolitano, Lucimar Pinheiro Rosseto

**Affiliations:** 1Campus Arthur Wesley Archibald, Evangelical University of Goiás, Anápolis 75083-515, Brazil; 2Campus Central, State University of Goiás, Anápolis 75132-400, Brazil

**Keywords:** Cerrado, ACANTHACEAE, plant biodiversity, phytochemistry, molecular modeling

## Abstract

The genus *Justicia* has more than 600 species distributed in both hemispheres, in the tropics and temperate regions, and it is used in the treatment of numerous pathologies. This study presents a review of the biological activities of plant extracts and isolated chemical constituents of *Justicia* (ACANTHACEAE), identified in the period from May 2011 to August 2022. We analyzed over 176 articles with various biological activities and chemical compound descriptions present in the 29 species of *Justicia*. These have a variety of applications, such as antioxidant and antimicrobial, with alkaloids and flavonoids (e.g., naringenin) the most frequently identified secondary metabolites. The most observed species were *Justicia gendarussa* Burm., *Justicia procumbens* L., *Justicia adhatoda* L., *Justicia spicigera* Schltdl, and *Justicia pectoralis* Jacq. The frontier molecular orbitals carried out using density functional theory (M062X and basis set 6-311++G(d,p) indicate reactive sites for naringenin compound and a chemical reaction on phytomedicine activity. The energy gap (206.99 kcal/mol) and dimer solid state packing point to chemical stability. Due to the wide variety of pharmacological uses of these species, this review points toward the development of new phytomedicines.

## 1. Introduction

The Cerrado is a highly heterogeneous landscape, and some of it is subject to severe threats and deserves special attention, including the Cerrado-Amazon transition, which coincides with an “arc of deforestation”, and rupestrian fields [1]. The region called Cerrado, located in the central portion of Brazilian territory, has changed abruptly in environmental, social and economic aspects. These changes were caused by the intense process of human occupation to which this ecosystem has been subjected, due to a sum of political interventions, natural features of the landscape and technological advances in agriculture [2]. The Cerrado is considered a Biodiversity Hotspot, which supports high species richness and thousands of endemic species [3]. Part of this huge biodiversity can be associated with the diversity of native vegetation types (e.g., grasslands, shrublands, typical savannas, and woodland savannas) that differ in grass cover, percentage of canopy cover, and dominant plant species, as well as fire dynamics and water availability [4]. Among the plant biodiversity, the ACANTHACEAE family has pantropical distribution, reaching some temperate areas, with approximately 240 genera and approximately 3250 species [5]. In Brazil, it is estimated that there are approximately 40 genera and 449 species, of which at least 254 are endemic, with a high concentration of species in the southeast and central-west regions [6].

The genus *Justicia* (Figure 1) comprises herbaceous plants or erect ascending shrubs with opposite leaves of crenate or entire maple. Terminal or axillary inflorescences with sessile or pedunculated flowers, solitary or cymosal in the axils of the bracts, are arranged in spikes or panicles. The 4-5-parted calyx and corolla are of varying colors (purple, red, lilac, white, yellow, or orange), lipped limbus, rear inner lip in pre-flowering, usually narrow, erect or curved. Sometimes concave, with entire apex, bifid or frontal lip slightly more bilobed, wider, more or less patent or curved, trilobed. Two stamens with filaments inserted near or above the middle of the filiform tube or slightly dilated at the base; anthers bitec, theca oblong, sometimes slightly curved or kidney-shaped. Two eggs in each locale. Capsule-like fruit with oblong, elliptical or obovate contour, with solid basal portion and laterally compressed and the upper part cylindrical, ovoid or subspherical portion [7].

The genus *Justicia* has more than 600 species widely distributed in both hemispheres, especially in the tropics, extending to temperate regions [7]. *Justicia* species are used in folk medicine for the treatment of numerous pathologies, such as depression, anemia, epilepsy, kidney infection, respiratory problems, gastrointestinal diseases, arthritis and fever [8]. Other biological activities have been determined from the plant extracts of *Justicia*, such as antioxidant [9,10,11,12,13,14,15,16,17,18,19,20,21], antimutagenic [18,19], anticancer [22,23,24,25], anti-HIV [26,27,28,29,30,31], antimicrobial [32,33,34,35,36,37,38,39,40,41,42], antidiabetic [43,44,45], among others. It is worth mentioning that there is currently relevant interest among research groups in evaluating a possible analgesic, anti-inflammatory and antiulcerogenic activity. A diversity of special metabolites is found in *Justicia*, mainly alkaloids [46,47,48,49,50,51], steroids [52,53,54], tannins [32,41,46,51], terpenoids [47,48,51,54,55,56,57,58], lignans [59,60,61,62,63,64] and flavonoids [16,20,21,22,23,24,31,32,51,65,66,67].

Many of the medicines currently available are derived from natural sources. In additional to their physiological roles in plants, flavonoids are important components of the human diet, although they are not considered nutrients. Flavonoids are an important class of plant secondary metabolites that serve several functions, including pigments and antioxidant activity. The biological activities from flavonoids make Cerrado plants good candidates for phytochemical studies, mainly naringenin, which belongs to the class of chalcones [68]. Naringenin is a naturally occurring flavonone (flavonoid) known to have a bioactive effect on human health, and it is found primarily in fruits (grapefruit and orange) and vegetables. Naringenin has several biological functions, such as antidiabetic, antiatherogenic, antidepressant, immunomodulatory, antitumor, anti-inflammatory, DNA protective, hypolipidemic, antioxidant, activator of peroxisome proliferator-activated receptors (PPARs) and memory enhancer. Several molecular mechanisms underlying their beneficial activities have been elucidated [69]. These have been found in several species of the genus *Justicia*, mainly *Justicia gendarussa* Burm, one of the species found in the Cerrado. This review evaluates the biological activities of plant extracts and chemical constituents of *Justicia* (ACANTHACEAE) in the period between May 2011 and August 2022.

## 2. Results and Discussion

### 2.1. Species, Compounds and Their Effects

We identified 29 species of the genus: *Justicia acuminatissima* (Miq.) Bremek, *Justicia adhatoda* L., *Justicia beddomei* (C.B.Clarcke) Bennet, *Justicia betonica* L., *Justicia brandegeeana* Wassh. & L.B. Sm., *Justicia carnea* Hook. Ex Nees, *Justicia extensa* T. Anderson, *Justicia flava* Vahl, *Justicia gangetica* L., *Justicia gendarussa* Burm, *Justicia graciliflora* (Standndl.) D.N. Gibson, *Justicia hypocrateriformis* Vahl, *Justicia insularis* T. Anderson, *Justicia neesii* Ramamoorthy, *Justicia nodicaulis* (Nees) Leonard, *Justicia paracambi* Braz., *Justicia pectoralis* Jacq., *Justicia procumbens* L., *Justicia refractifolia* (Kuntze) Leonard, *Justicia schimperiana* T. Anderson, *Justicia secunda* Vahl, *Justicia simplex* D. Don., *Justicia spicigera* Schldtl, *Justicia subsessilis* Oliv., *Justicia thunbergioides* (Lindau) Leonard, *Justicia tranquebariensis* L., *Justicia vahlii* Roth., *Justicia wasshauseniana* Profice and *Justicia wynaadensis* B Heyne. All scientific names were verified in the International Plant Name Index (IPNI). The most used plant parts were leaves, with 121 citations, followed by the aerial parts, with 29 citations, and the whole plant, with 18 citations. These values can be explained by the ease of harvesting, since the leaves are available most of the year [70].

There was a prevalence of studies carried out in India, followed by Brazil, Nigeria, and China (49, 26, 21 and 15, respectively). This is due to the fact that India has many renowned universities and institutions with a growing faculty in research, as well as investment in innovation [71]. In Table 1, the species of *Justicia*, parts and crude extracts used were compiled, in addition to studies related to the presence or absence of chemical and biological information. The most representative biological activities were: antioxidant, with 30 citations, and antimicrobial, with 23 citations. Methanolic extracts from the species *J. adhatoda* L., *J. beddomei* (C.B.Clarcke) Bennet, *J. brandegeeana* Wassh. & L.B. Sm., *J. gendarussa* Burm., *J. pectoralis* Jacq. and *J. thunbergioides* (Lindau) Leonard, showed antioxidant effects. Studies about the antimicrobial effects were more representative with the ethanolic extract of the species *J. acuminatissima* (Miq.) Bremek. (*Staphylococcus aureus*, *Bacillus cereus*, *Escherichia coli*, *Salmonella typhimurium* and *Candida albicans*), *J. gendarussa* Burm. (*S. aureus*, *B. subtilis*, *E. coli* and *Klebsiella pneumoniae*), *J. pectoralis* Jacq. (*S. aureus* and *S. epidermidis*) and *J. simplex* D. Don. (*S. aureus*, *K. pneumoniae*, *E. coli* and *Pseudomonas aeruginosa*).

Several compounds isolated from *Justicia* showed biological activities (Table 2). Steroids 1 (glycosylated β-sitosterol) and 2 glycosylated stigmasterol) isolated from the ethanolic extract of the aerial parts of *J. acuminatissima* (Miq.) Bremek., showed reduction of inflammatory infiltrates and edema, small to moderate injury lesion, 24 h after treatment, used topically after the administration of a gel containing the extract during the therapeutic ultrasound session, after an injury caused by the free fall of a weight of 300 g at a height of 30 cm on the calf of rats, showing a significant reduction in paw edema in rats. Terpene 3 (phytol), extracted and isolated from the leaves of *J. gendarussa* Burm., showed potent inflammatory inhibition (68.03%) when compared to standard dicolfenac (5 mg/Kg). Compound 4 (apigenin), an alkaloid, showed an anti-inflammatory effect through the TLR-NF-κB signaling pathway (Toll-like receptors linked to transcription factors) using hPBMCs (Peripheral Blood Mononuclear Cells) induced by LDL-ox (Oxidized Lipoprotein) in an in vitro model, reducing the release of TLR-4, receptors that stimulate the production of pro-inflammatory mediators. In addition, compounds 5 (naringenin) and 6 (kaempferol), from the methanolic extract of the roots of the same species, showed cytotoxic effects against human cancer cell lines: HT-29 (19 and 6 µg/mL), HeLa (15 and 5 µg/mL) and BxPC-3 (57 and 23 µg/mL) inhibiting their growths, respectively.

Flavonoid 7 (3,3′,4′-trihydroxyflavone) showed antimicrobial activity, maximum zone of inhibition, from the methanolic extract of leaves of *J. wynaadensis* B. Heyne, against *Enterocytes faecalis* (19 mm) and MIC = 32 µg/mL, *S. aureus* (18 mm) and MIC = 32 µg/mL, *E. coli* (17 mm) and MIC = 128 µg/mL, *Enterobacter aerogenes* (18 mm) and MIC = 128 µg/mL, *S. epidermidis* (11 mm) and *K. pneumoniae* (17 mm) and MIC = 64 µg/mL, from wounds of diabetics with urinary tract infection, compared to the standard chloramphenicol (19 mm) and MIC = 1024 µg/mL. The same activity showed by alkaloids 8 (vasicoline) (has greater inhibitory capacity in the biosynthesis of fatty acids and stops the activity of the mtFabH enzyme of *Mycobacterium tuberculosis*, being able to interrupt the infection in its initial stage) and 9 (vasicine) [inhibiting the growth of *K. pneumoniae* (10.2 mm) and MIC = 6.25 µg/mL, *E. coli* (12.5 mm) and MIC = 3.125 µg/mL, *P. aeruginosa* (6 mm), *S. pyogenes* (9.8 mm) and MIC = 25 µg/mL, *S. aureus* (12.8 mm) and MIC = 12.5 µg/mL, *S. marcescens* (8.2 mm) and MIC = 3.125 µg/mL, when compared to ofloxacin (8.8 mm, 9.1 mm, 2 mm, 9.5 mm and 7.8 mm, respectively) and *A. flavus* (10.5) and MIC = 3.125 µg/mL, *C. albicans* (14.2) and MIC = 12.5 µg/mL and *C. neoformans* (11.5 mm) and MIC = 25 µg/mL when compared with amphotericin (12 mm, 11 mm and 10 mm, respectively)], obtained from the extract of leaves of *J. adhatoda* L. Compound 9 (vasicine) also showed antioxidant effects (protecting deoxyribose from the action of free radicals with IC_50_ 539.64 µg/mL and having a strong chelating activity) and anticancer effects [inhibitory effect on the growth of prostate cancer cells (IC_50_ 81.11 µg/mL)].

Etamine (10), a nitrogen compound, from the ethanolic extract of the leaves of *J. gendarussa* Burm., exhibited DPPH radical scavenging activity with IC_50_ = 22.55 µM, and Quercetin as positive control, IC_50_ = 18.56 µM. Pyrrolidines 11 (Secundallerone B) and 12 (Secundallerone C), along with acid 13 (2-caffeoyloxy-4-hydroxy-glutaric acid) showed antidiabetic effects, such as α-glucosidase inhibitors when extracted from leaves of *J. secunda* Vahl., using the methanolic extract. There are compounds that have various biological activities. An example is kaepferitrin (14), an alkaloid that has shown antinociceptive, cytotoxic effects against cancer cells (against human cervical carcinoma cells, inducing apoptosis of these cells by 35% and inhibiting their growth by 53%), antidiabetic and anticonvulsant. Another alkaloid 15 (gendarussin A), isolated from the ethanolic extract of the leaves of *J. gendarussa* Burm., has an anti-HIV cytotoxic effect, decreasing viral load, increasing anti-HIV activity (reverse transcriptase inhibition), with an IC_50_ value of 235.3 ppm.

The isolated compounds that presented the highest frequency of published works were lignans. There were 38 lignans studied and surveyed, with different biological activities evidenced. Lignans 16 (6′-hydroxyl justicidin A), 17 (6′-hydroxyl justicidin B), 18 (6′-hydroxyl justicidin C), 19 (Justicidin A), 20 (Chinensinaphthol methyl ether), 21 (Taiwanin E methyl ether), 22 (Paclitaxel) and 23 (Podophyllotoxin) showed cytotoxic effects against cancer cells K652 (leukemia) and TSGH8301 (bladder carcinoma), with IC_50_ = 0.148 µM, IC_50_ = 2.356 µM, IC_50_ = 15.2 µM, IC_50_ = 1 µM, IC_50_ = 106.2 µM, IC_50_ = 100 µM, IC_50_ = 48 µM and IC_50_ = 28.5 µM, respectively, and lignans 16, 17, 18, 20, 21, 22 and 23 from the ethanolic extracts and 19 from the methanolic extract of the species *J. procumbens* L. On the other hand, lignans 19 and 24 (Justicidin B), 25 (Justicidin C) and 26 (Phyllamyricin C) showed anti-inflammatory and anti-allergic effects, inhibiting the infiltration of inflammatory cells in the airways of rats, to the point of decreasing bronchoconstriction, reducing the levels of IgE (91.7%), IL-4 (39.2%), IL-5 (51.7%) and eotaxin (66.5%) with values of IC_50_ = 0.5 µM to compound 24 and IC_50_ = 5 µM to compounds 25 and 26. Compounds 25 and 27 (Pronaphthalide A) showed cytotoxic activities (significant effect on cell viability, affecting the methylation, deoxidation, and glycosylation activity of BGC-823 cancer cells). In addition to these, other lignans 20, 28 (Procumbenoside J), 29 (Tuberculatin) and 30 (Diphyllin) showed suggestive effects on cytotoxicity against BGC-823 cancer cells (gastric carcinoma) with value of IC_50_ = 0.135 µM to compound 27 and compound 31 (Procumbenoside H) against colon cancer cells, derived from ethanolic extracts of the same species, with value of IC_50_ = 17.908 µM.

Compounds 30 and 41 (Justicianene D) showed cytotoxic activity against cervical (30), A549 and H460 (30), breast (41) and lung cancer cells, with value of IC_50_ = 90 µM, derived from the ethanolic extract of the same species. Other compounds derived from the ethanolic extract of the leaves of the species *J. gendarussa* Burm., (+)-pinoresinol (32), a lignin, exhibited DPPH radical scavenging activity with value of IC_50_ = 28.61 µM. In addition, two compounds, 33 (2’-methoxy-4”-hydroxydimetoxykobusin) and 34 (Brazoide A), showed anti-inflammatory activity in macrophages with IC_50_ values of 20.95 and 16.5 µM, respectively, compared to dexamethasone as a control (11.69 µM). Other lignans 35 (Justiprocumin A) and 36 (Justiprocumin B) showed cytotoxic effects against HIV viruses (reverse transcriptase inhibition), with IC_50_ values between 14 and 21 nM compared to AZT (Zidovudine) with IC_50_ between 77 and 95 nM, coming from the methanolic extract of the roots and stems of the same species.

Compound 37 (Pateniflorin A) also showed anti-HIV activity with IC_50_ = 26.9 nM, from methanolic extract of the stems and roots of the *J. gendarussa* Burm. Compound 38 (Triacontanoic ester of 5-hydroxyjustisolin), another lignin, showed no toxicity to the animals (rats) tested, increasing their survival capacity when induced to the tumor [mammary (MDA MB-231) and cervical carcinoma (HeLA), from two extracts (petroleum ether and ethanol) from aerial parts of *J. simplex* D. Don., with values of IC_50_ = 15.15 and IC_50_ = 11.852 µg/mL, respectively. Two terpenoid compounds identified and isolated from the methanolic extract of leaves of *J. insuaris* T. Anderson, 39 (16(α/β)-hydroxy-cleroda-3,13 (14)*Z*-dien-15,16-olide) and 40 (16-oxo-cleroda-3,13(14)E-dien-15-oic acid), showed cytotoxic activity against ovarian cancer cells (OVCAR-4 and OVCAR-8), inducing apoptosis with values of IC_50_ = 5.7 and IC_50_ = 16.6 µM, respectively, to compound 39 and IC_50_ = 4.4 and IC_50_ = 11.8 µM, respectively to compound 40. Finally, six compounds identified and isolated from the ethyl acetate extract of the aerial parts of *J. spicigera* Schltdl. Inhibited the activity of the enzyme tyrosine phosphatase B, a key regulator of insulin signaling cascades, evidencing a synergistic effect of all six compounds, namely 42 [2-*N*-(p-coumaroyl)-3H-phenoxazin-3-one, IC_50_ = 159.1 µM], 43 (3″-O-acetyl-kaempferitrin, IC_50_ = 306.7 µM), 14 (Kaempferitrin, IC_50_ = 306.7 µM, 44 (kaempferol 7-O-α-L-rhamnopyranoside), 45 (perisbivalvine B, IC_50_ = 106.6 µM) and 46 (2,5-dimethoxy-p-benzoquinone, IC_50_ = 455.5 µM). This was the first report of the presence of phenoxazines in the genus *Justicia*. In this paper, 46 compounds (Figure 2) were identified and isolated from species of the genus *Justicia*. 

### 2.2. Molecular Modeling of Naringenin

Compound 5, 2,3-dihydro-5,7-dihydroxy-2-(4-hydroxy-phenyl-4H-1-benzopyran-4-one, (racemic naringenin) (Figure 3) crystallizes in the P2_1_/c monoclinic space group, and the crystal data and refinement details are summarized in Table 3. The asymmetric unit is shown in the ORTEP (Oak Ridge Thermal Ellipsoid Plot) diagram in which the angle between the mean plane of the benzopyrone ring and hydroxyphenyl ring is approximately perpendicular with the value of 85.73°. The pyrone ring appears as half chair conformation confirmed by the ring-puckering parameters Q = 0.4215 Å and ϕ = 246.8°, as described by Cremer and Pople [191]. In addition, the hydroxyphenyl ring (C10) is bonded equatorially to this pyrone ring and its dihedral angles O1-C1-C1O-C11, C2-C1-C10-C11, C2-C1-C10-C15, and O1-C1-C10-C15 are 120.51°, −115.58°, 62.47° and −61.44°, respectively, as shown in Table 4.

The crystal structure of naringenin makes a conjugated six-membered ring, forming strong O3–H3···O2 intramolecular interactions, as shown in Table 5. The crystal packing for naringenin is formed by dimers, which are responsible for generating O4–H4···O5 intermolecular interactions, which can be described as $R_{2}^{2}$ (24) [192] (Figure 4a). In a two-dimensional hydrogen-bonding arrangement, there is a chain appearing in a zigzag and growing along the *c*-axis, which is formed by the O5–H5···O2 intermolecular interactions and can be described as $C_{\;1}^{1}$ (9) (Figure 4b). Additionally, the C15–H15···O4 intermolecular interactions also form a zigzag chain, which grows along the *b*-axis and can be described as $C_{1}^{1}$ (10) (Figure 4c). The crystal packing is formed by the dimers (involving hydroxyl groups), and the zigzag chains, which generate a two-dimensional crystalline network, as shown in Figure 4d.

We employed HS mapped over *d*_norm_. (ranging from −0.679 to 1.270 Å) analysis to interpret the most dominant interactions responsible for crystal packing, as shown in Figure 5. These interactions are analyzed based on the distances between the internal nucleus of the HS within the molecule (*d*_i_) and the external nucleus of the HS within the molecule (*d*_e_), where the red dots represent the strong interactions. For naringenin, the red dots in Figure 5a correspond to a dimer formed by O4–H4···O5 intermolecular interaction. In addition, the red dots in Figure 5b are related to the O5–H5···O2 intermolecular interaction. Finally, the non-classical C15–H15···O4 intermolecular interaction is represented by the red dots on the HS, as shown in Figure 5c.

The 2D fingerprint plot of naringenin is shown in Figure 6. The 2D fingerprint plots (*d_i_* vs. *d_e_*) quantify the types of intermolecular contacts in the solid-state arrangement [193]. These H···H contacts make up 35.0% of the HS of naringenin because it is an organic compound [194]. The red spots represent O···H/H···O contacts, which are the second largest contributions, with 31.8% of the HS of naringenin, and it is shown as the spikes at the bottom of the 2D fingerprint plot. Finally, C···H/H···C contacts represent 23.5% of the HS of naringenin.

Naringenin has a molecular weight of 272.257 g/mol, resulting from the addition of three hydroxyl groups 4′, 5 and 7 carbons in the backbone of flavonoids, and its molecular formula is C_15_H_12_O_5_ [195,196]. This compound is found in high concentrations, especially in grapefruit (43.5 mg/100 mL), followed by orange juice (2.13 mg/100 mL) and lemon juice (0.38 mg/100 mL) [197]. Naringenin has a range of biological effects on human health, which include a reduction in lipid peroxidation markers, defense of metabolism, increase in antioxidants, reduction of reactive carbohydrate species, as well as modulation of the immune response [198,199]. In vitro and in vivo animal studies have reinforced evidence of the diversity of pharmacological effects of naringenin; among them, we highlight hepatoprotective, antiatherogenic, anti-inflammatory, antimutagenic, anticancer and antimicrobial activity [200]. Although we have identified in the literature that there is an enormous amount of data on the in vitro biological effects of naringenin, there are still few studies available on its therapeutic potential [201], and thus, further clinical studies are needed, aiming at the safety, efficacy and bioavailability of naringenin in humans.

The frontier molecular orbitals (FMO) taken from the natural bond orbital (NBO) analysis for compound 5 (naringenin) were carried out at the M062X/6-311+G(d,p) level of theory, and this is shown in Figure 7. The HOMO appears as a π bonding orbital, and it is localized on the phenyl π bonding region, which is characteristic of the nucleophilic region with an energy value of −194.44 kcal/mol. The LUMO orbital appears as a π antibonding orbital, and it is localized on the π region of the pyrone ring with an energy value of 12.55 kcal/mol. The energy gap (206.99 kcal/mol) shows that compound 5 (naringenin) is chemically stable.

The MEP is a physicochemical tool that helps to predict the reactive sites to be targeted in a chemical reaction and gives information about molecular interactions. The electrostatic potential at a given point ρ(r) in the vicinity of a molecule can be calculated by Equation (1).
(1)$$V\left(r\right)=\underset{\alpha }{\sum }\frac{Z_{\alpha }}{\left|r-R_{\alpha }\right|}-\int \frac{\rho \left(r^{\prime }\right)}{\left|r-r^{\prime }\right|}dr^{\prime }$$
where V(r) is the potential energy by a positive unit charge at point r; $Z_{\alpha }$ is the nuclear charge of the atom α located at position $R_{\alpha }$, and $\rho \left(r^{\prime }\right)$ is the electron density. The tridimensional molecular electrostatic potential (3D-MEP) representation for compound 5 (naringenin) shows that the oxygen atom of the carbonyl group localizes the most negative region (red), with the value of −26.85 kcal/mol (Figure 8). On the other hand, the positive region (blue) is around the hydroxyl hydrogen atom with a value of 45.11 kcal/mol. In conclusion, due to the presence of interactions within the hydroxyl group O4–H4···O5 in the crystal structures, we can assume a nucleophilic attack within this hydroxyl region.

The root of the mean squared (RMS) value between experimental geometries and theoretical calculation was 0.0135, predicted by Mercury software. The overlapping of the X-ray (black) and M062X/6-311+G(d,p) level of theory (green) is shown in Figure 9a. The comparative graphs for the bond lengths and angles obtained for experimental geometries and theoretical calculation are shown in Figure 9b,c. The mean absolute percentage deviations (MAPD) were calculated and defined by Equation (2):(2)$$MAPD=\frac{100}{n}\overset{n}{\underset{i=1}{\sum }}\left|\frac{\chi_{XRD}-\chi_{DFT}}{\chi_{XRD}}\right|.$$
where $\chi_{XRD}$ and $\chi_{DFT}$ represents the geometric parameters for experimental geometries and theoretical calculation data, respectively. The MAPD values for bond lengths and angles were 0.86 and 0.64 for experimental geometries and theoretical calculation data of naringenin. The R^2^ values for bond lengths were 0.9771 and 0.9670, for experimental geometries and theoretical calculation data of naringenin, respectively.

The conformation analysis for naringenin was performed by Ávila and coworkers [202] showing two stable conformers (conformer 1 and conformer 2) obtained by molecular dynamics simulation in a DMSO solution. The conformation found in the solid state is approximate to conformer 2. Conformer 2 has the phenol ring in an equatorial position and it is 2.39 kcal/mol more stable than conformer 1. In addition, the free energy barrier is 3.75 kcal/mol for converting the conformer 1 to conformer 2 direct process and 6.15 kcal/mol for the reverse process, so a suggested conformation equilibrium can occur in the DMSO solution at 298.15 K.

## 3. Method

### 3.1. Systematic Review

The present study was carried out through a systematic review of articles, dissertations and theses published between May 2011 and August 2022. The searched electronic databases were ISI Web of Science and Scholar Google, using the following keywords: ACANTHACEAE, *Justicia* and Medicinal plants. The collected data were screened by analyzing titles, keywords, abstract and full texts. The literature containing information on isolation and property of different phytochemical compounds from species of the genus *Justicia* were included, too. More than 6500 articles, dissertations and theses were found on databases. Figure 10 shows the search and selection processes.

### 3.2. Molecular Modeling Analysis

The (R,S)-naringenin structure was extracted from the Cambridge Crystallography Data Centre (CCDC) with the code 1143928. Platon (2009) [203] and Mercury (2020) [204] were followed to analyze and draw the crystal supramolecular arrangement. Hirshfeld surface analysis (HS) (2009) [205] is a useful tool to understand the intermolecular contacts among atoms and crystal packing. HS is calculated based on the distances between the internal nucleus of the HS within the molecule (d_i_) and the external nucleus of the HS within the molecule (d_e_) [206]. The normalized contact distance (d_norm_), which combines the normalized de and d_i_ with the van der Waals radius, is used to identify the most important contacts present in the molecule. Moreover, the 2D fingerprint plots provide the frequency and quantitative information about the calculated intermolecular contacts. For this purpose, we used Crystal Explorer 21.5 [207] software to generate this HS surface and to calculate the 2D fingerprint plots. The electronic structure calculations were carried out with the Gaussian 16 [207] program package for compound 5 (naringenin). Full geometry optimization was carried out using density functional theory (DFT), with exchange-correlation functional M062X and basis set 6-311++G(d,p) [206], and the electronic properties, such as the highest occupied molecular orbital (HOMO), the lowest unoccupied molecular orbital (LUMO) and the molecular electrostatic potential (MEP), were calculated [207].

## 4. Conclusions

There were 29 species of the genus *Justicia* studied all of which presented information regarding chemical information, with 28 biological activities presented: 19 had their compounds identified, and 10 species had their compounds isolated. Alkaloids and flavonoids (e.g., naringenin) were the compounds of the active extracts that had the highest frequency of identification among the researched data. The secondary metabolites that most frequently showed biological effects were lignans. The most researched species were *Justicia gendarussa* Burm, *Justicia adhatoda* L., *Justicia procubens* L., *Justicia spicigera* Schltdl, and *Justicia secunda* Vahl., with frequency values of articles surveyed of 40, 20, 19, 18 and 16, respectively. Species of the genus *Justicia* have a range of biological uses, identified as antioxidant, antimicrobial and anticancer, among others. The first two are the most representative; however, we would suggest the need for further research. The FMO taken from NBO analysis indicates reactive sites for compound 5 (naringenin) to be targeted in a chemical reaction on phytomedical activity. The energy gap (206.99 kcal/mol) and dimer solid state packing ($R_{2}^{2}$ (24) symmetry) indicates that naringenin is chemically stable.

## Figures and Tables

**Figure 1 molecules-28-01190-f001:**
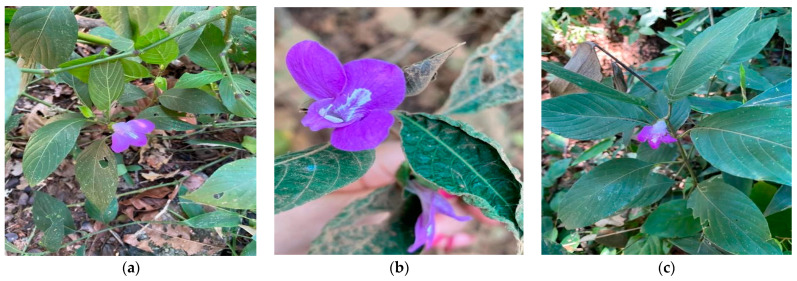
Specimens of *Justicia thunbergioides* (Lindau) Leonard. (**a**) Flower and habit in Cocalzinho de Goias (−15°44′47′′; −48°44′47′′). (**b**) Flower and habit in Central Plateau Protection Area (Fercal) Federal District (−15°30′49′′; −47°57′56′′). (**c**) Flower and habit in Onofre Quinan Park-Anapolis Goias (−16°20′22′′; −48°57′49′′).

**Figure 2 molecules-28-01190-f002:**
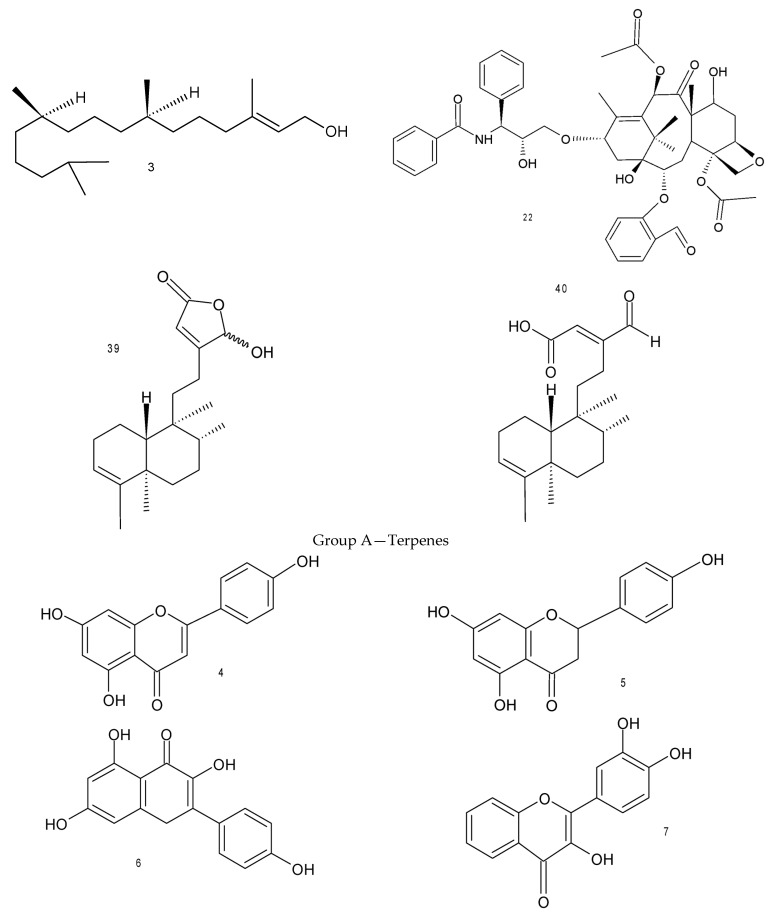
Compounds isolated from species of *Justicia*.

**Figure 3 molecules-28-01190-f003:**
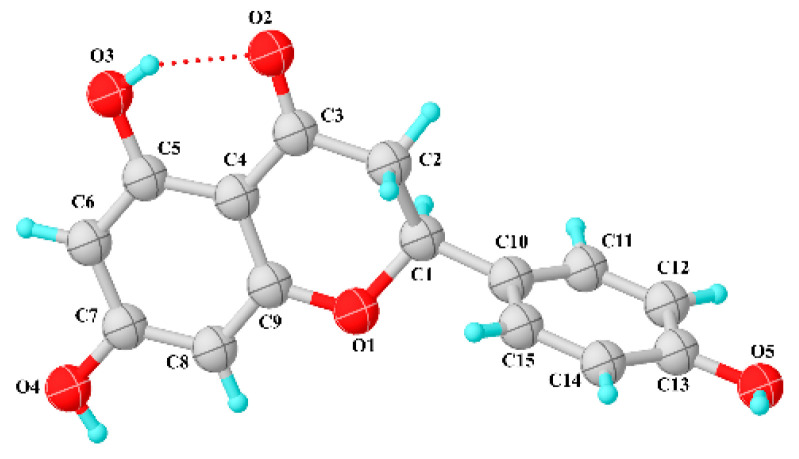
ORTEP representation of the asymmetric unit for naringenin with the atom numbering scheme. Ellipsoids are drawn at the 50% probability level.

**Figure 4 molecules-28-01190-f004:**
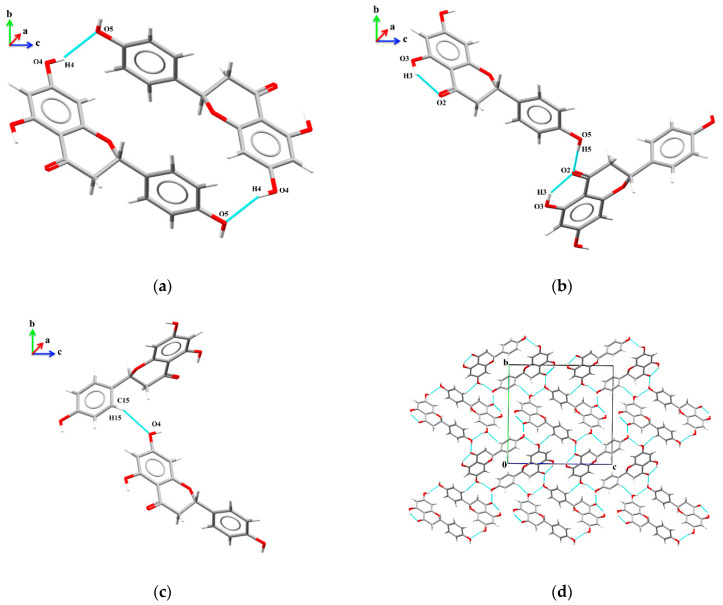
Representation of interactions responsible for naringenin crystal packing. These interactions are (**a**) O4–H4···O5; (**b**) O5–H5···O2; (**c**) C15–H15···O4 and (**d**) two-dimensional crystal packing.

**Figure 5 molecules-28-01190-f005:**
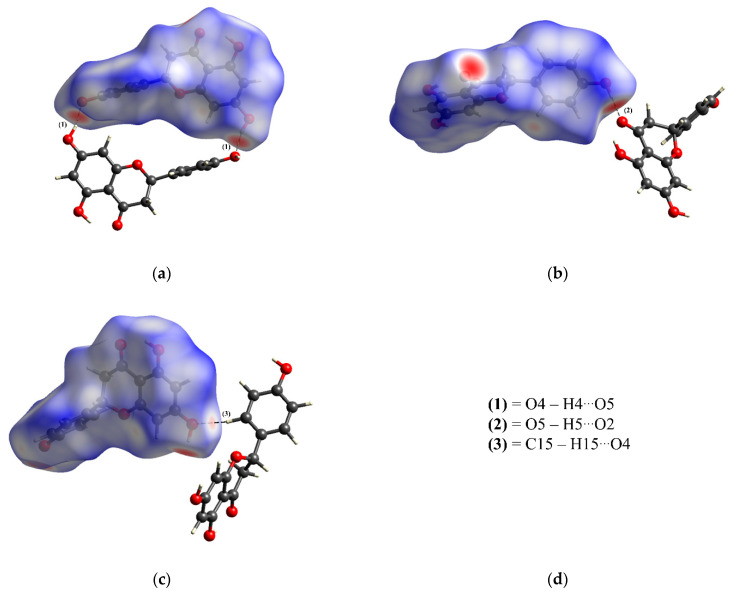
The HS d_norm_ showing (**a**) O4–H4···O5; (**b**) O5–H5···O2 and (**c**) C15–H15···O4 intermolecular interactions found in the two-dimensional crystal packing for naringenin (**d**) types of the intermolecular interactions found in the HS for naringenin. The red dots represent the strong contacts.

**Figure 6 molecules-28-01190-f006:**
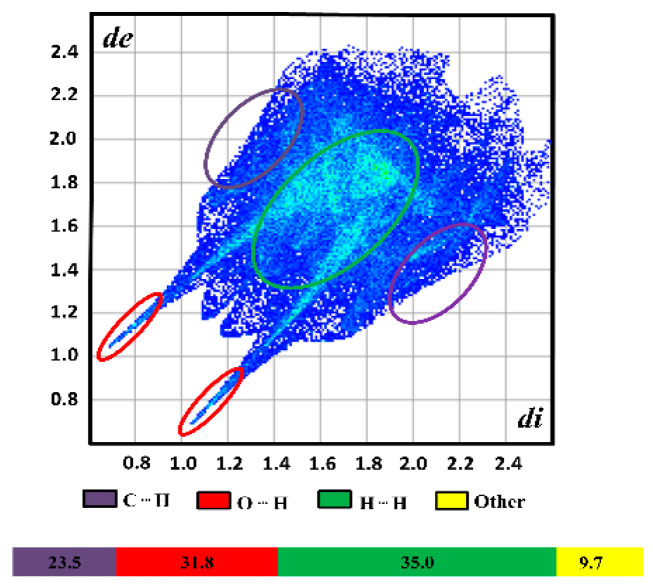
The 2D fingerprint plot representation of naringenin. The reciprocal contacts were included.

**Figure 7 molecules-28-01190-f007:**
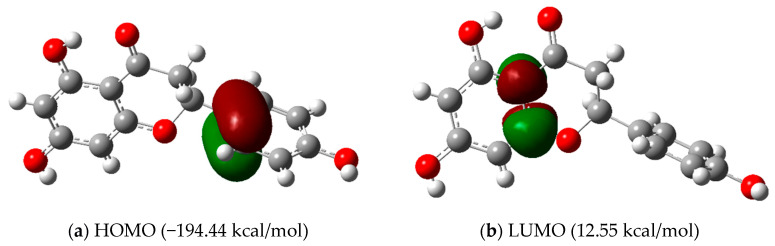
NBO orbital analysis with the isovalue of 0.05 atomic units, showing (**a**) the HOMO and (**b**) the LUMO orbitals, which are the π-bonding orbital and π-antibonding orbital, respectively.

**Figure 8 molecules-28-01190-f008:**
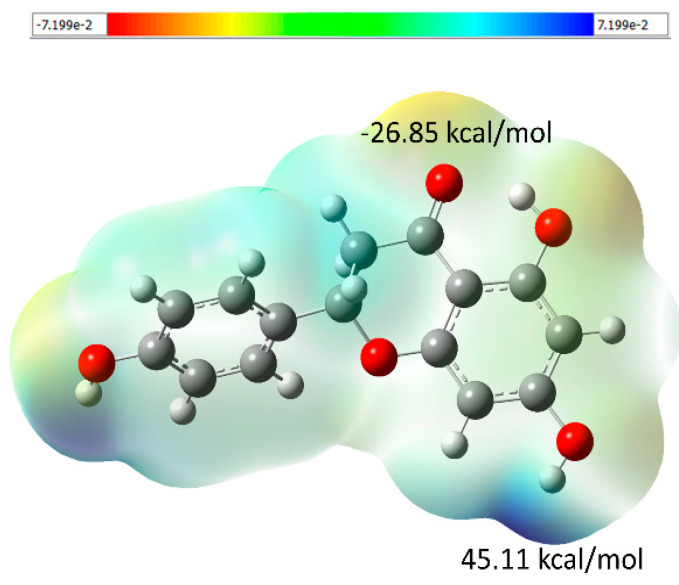
The molecular electrostatic potential (MEP) surface mapped for compound 5 (naringenin) shows the region rich in electrons (red-colored) and the region depleted in electrons (blue-colored). The density isovalue of 4.0 × 10^−4^ electrons/bohr^3^ was used to generate the molecular electrostatic potential surfaces.

**Figure 9 molecules-28-01190-f009:**
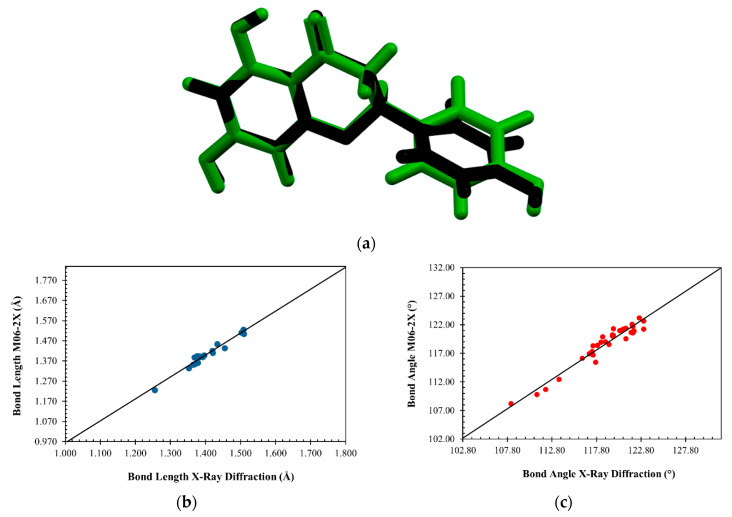
Overlapping of M062X/6-311+G (d,p) level of theory (green) and experimental data (black) structure of naringenin (**a**), comparison graphs of the geometric bond length and angle, obtained by experimental and theoretical calculation (**b**,**c**).

**Figure 10 molecules-28-01190-f010:**
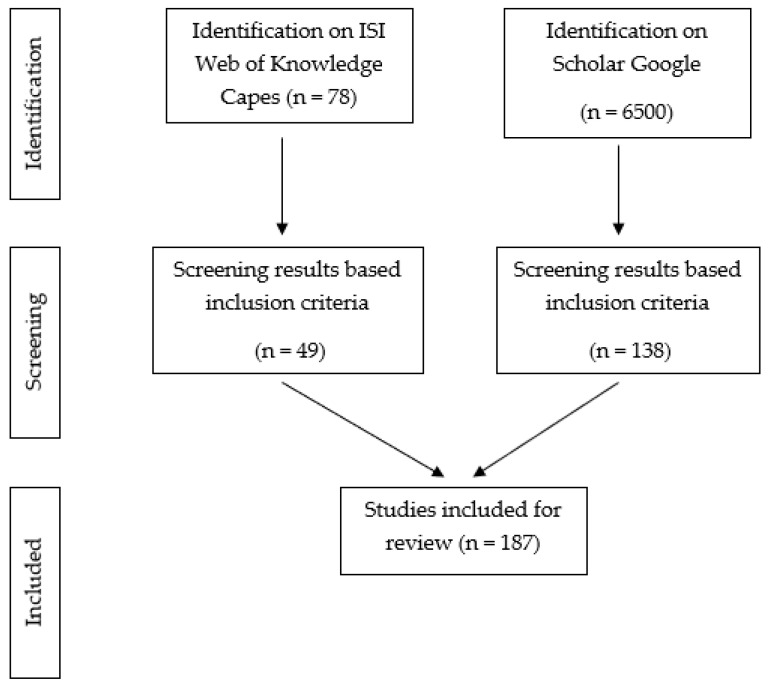
The flow chart of the identification and selection process.

**Table 1 molecules-28-01190-t001:** Information on the survey of species of the genus *Justicia*, parts and crude extracts used, chemical information, biological information and geographic distribution.

Species	Tested Part	Extract	Chemical Information	Biological Information	Origin	Reference
*Justicia acuminatissima* (Miq.) Bremek	Aerial parts ^a,c^	Ethanol ^a,c^	Yes ^a,c,d^	Yes ^a,b,c,d,e^	Brazil ^a,b,c,d^	Corrêa et al., 2014 ^a^ [33]
	NI ^b^	NI ^b^	No ^b,e^			Cordeiro et al., 2019 ^b^ [11]
	Leaf ^d,e^	Aqueous ^d,e^				Corrêa, 2013 ^c^ [25]
						Corrêa et al., 2014 ^b,d^ [34]
						Verdam et al., 2015 ^e^ [12]
*Justicia adhatoda* L.	Leaf ^a,c,d,e,f,g,i,j,k,l,m,n,p,q,r,s,t^	Ethanol ^a,d,f,f,j,k,l,m,q,s^	Yes ^b,c,e,f,h,i,j,k,l,p,r,t^	Yes ^a,b,c,d,e,f,g,h,i,jk,l,n,o,p,q,r,s,t^	India ^a,b,c,e,f,g,h,i,j,l,o,q,r^	Kaur; Kaur; Arora, 2015 ^a^ [9]
	Root ^l^	Methanol ^c,e,i,j,n,t^	No ^d,e,i,o,p,q,s^	No ^n^	Armenia ^d^	Chaliha et al., 2016 ^b^ [35]
	NI ^k^	Ethyl acetate ^d,r^			Pakistan ^k,m,s,t^	Pa; Mathew, 2012 ^c^ [36]
		NI ^h^			Sri Lanka ^n^	Barth et al., 2015 ^d^ [72]
		Ether ^i^			Bangladesh ^p^	Jha et al., 2012 ^e^ [37]
		Chloroform ^j,q^				Jha et al., 2014 ^f^ [13]
		Acetone ^i^				Kaur et al., 2016 ^g^ [9]
		Aqueous ^i,k,p,q,r^				Gutti et al., 2018 ^h^ [73]
		Hexane ^j^				Vinunkonda et al., 2012 ^i^ [38]
		Butanol ^j^				Dhankhar et al., 2014 ^j^ [17]
						Rasheed et al., 2013 ^k^ [45]
						Abhishek; Apurva; Joshi, 2014 ^l^ [74]
						Aziz et al., 2017 ^m^ [37]
						Someya et al., 2018 ^n^ [75]
						Thanigaivel et al., 2017 ^o^ [76]
						Chowdhury et al., 2020 ^p^ [77]
						Saran et al., 2019 ^q^ [18]
						Sudevan et al., 2019 ^r^ [21]
						Ameer et al., 2021 ^s^ [43]
						Basit et al., 2022 ^t^ [78]
*Justicia beddomei* (C.B. Clarke) Bennet	Leaf ^a^	Ethyl acetate ^a,c^	Yes ^a,c^	Yes ^a,b,c^	India ^a,b,c^	Prabavathy; Valli Nachiyar, 2013 ^a^ [79]
	Aerial parts ^b,c^	Methanol ^b,c^	No ^b^			
		Ether ^c^				Marathakam; Kannappan; Santhiagu, 2014 ^b^ [44]
		Chloroform ^c^				
						Marathakam et al., 2012 ^c^ [14]
*Justicia betonica* L.	Leaf ^a^	Ether ^a^	Yes ^a,b^	Yes ^a,b^	Uganda ^a^	Bbosa et al., 2013 ^a^ [46]
	Whole plant ^b^	Methanolic ^b^			India ^b^	Naik et al., 2022 ^b^ [32]
		Aqueous ^b^				
		Acetone ^b^				
		Ethanolic ^b^				
		n-hexane ^b^				
*Justicia brandegeeana* Wassh. & L.B. Sm.	Leaf ^a^	Methanol ^a^	Yes ^a,b^	Yes ^a^	Brazil ^a^	Cassola et al., 2019 ^a^ [19]
	Aerial parts ^b^	NI ^b^		No ^b^	China ^b^	Jiang et al., 2014 ^b^ [80]
*Justicia carnea* Hook. Ex Nees	Leaf ^a,b,c,d,e^	Ethanol ^a,b,d^	Yes ^a,b,c,e^	Yes ^b,c,d^	Nigeria ^a,b,c,d,e^	Otuokere et al., 2016 ^a^ [81]
		Aqueous ^c^	No ^d^	No ^a,e^		Onyeabo et al., 2017 ^b^ [82]
		Hexane ^e^				Anthonia et al., 2019 ^c^ [58]
						Akintimehin et al., 2021 ^d^ [83]
						Ajuru et al., 2022 ^e^ [84]
*Justicia extensa* T. Anderson	Leaf ^a^	Methanol ^a^	No ^a^	Yes ^a^	Nigeria ^a^	Sowemimo; Adio; Fageyinbo, 2011 ^a^ [85]

*Justicia flava* Vahl	Leaf ^a,b,c,d,e^	Methanol ^a,b,c,d,e^	Yes ^a,b^	Yes ^a,b,c,d,e,f,g^	Ghana ^a^	Agyare et al., 2013 ^a^ [31]
	Aerial parts ^f^	Aqueous ^f,g^	No ^c,d,e,f,g^		Nigeria ^b,c,d,e^	Bafor et al., 2019 ^b^ [49]
	Whole plant ^g^				Ivory Coast ^f,g^	Bafor; Prendergast; Wray, 2020 ^c^ [86]

						Bafor et al., 2019 ^d^ [59]
						Bafor et al., 2019 ^e^ [87]
						Wenceslas et al., 2021 ^f^ [88]
						Kounamé et al., 2021 ^g^ [89]
*Justicia gangetica* L.	Leaf ^a^	Ethyl acetate ^a^	Yes ^a^	Yes ^a^	Thailand ^a^	Stewart et al., 2013 ^a^ [90]
*Justicia gendarussa* Burm	Stem ^a,d,q,t,w,i′^	Methanol ^a,b,d,e,f,h,i,k,l,m,p,q,s,t,u,v,b′,h′,i′,j′^	Yes ^a,b,c,e,h,i,l,p,q,r,s,t,u,v,x,w,y,z,a′,b′,c′^	Yes ^a,b,c,d,e,f,g,h,i,j,k,l,m,n,o,r,s,t,x,y,z^	Vietnam ^a,i′^	Zhang et al., 2017 ^a^ [25]
	Root ^a,f,g,q,s,b′,c′,i′^	Aqueous ^b,c,b′,c′,f′,o′^	Yes ^d′,f′,g′,h′,i′,j′,l′,m′,n′,o′^	Yes ^a′,b′,c′,d′,e′,f′,g′,h′,i′,k′,n′,o′^	Malaysia ^b,h,p,e′^	Ayob et al., 2014 ^b^ [24]
	Leaf ^b,c,e,g,h,i,l,m,n,o,p,q,r,t,u,v,x,y,z^	Ethanol ^c,j,n,o,t,x,y,z,c′,d′,e′,f′,g′,k′,l′,m′,n′,o′^	No ^d,f,g,j,k,m,n,o,e′,k′^	No ^p,q,u,v,w,j′,l′,m′^	India ^c,d,f,g,i,j,m,n,o,r,s,t,w,z,b′,k′,l′,o′^	Subramanian; Jothimanivannan; Moorthy, 2012 ^c^ [40]
	Leaf ^b′,d′,e′,f′,g′,h′,j′,k′,l′,m′,o′^	Hydroalcoholic ^g^			Brazil ^e,x^	
	Whole plant ^d^	Ethyl acetate ^i,n,o′^			Bangladesh ^k,l^	Sugumaran et al., 2013 ^d^ [41]
	Aerial parts ^j,k,n′^	Chloroform ^i^			Indonesia ^q,u,v,y,a′,c′,d′,f′,g′,h′,j′,m′^	Cassola et al., 2019 ^e^ [19]
	NI ^a′^	Ether ^o,t,w,b′^			China ^n′^	Kumar et al., 2012 ^f^ [91]
		NI ^r,a′^				Patel; Zaveri, 2012 ^g^ [50]
		n-hexane ^o′^				Ayob; Samad; Bohari, 2013 ^h^ [23]
						Kowsalya; Sankaranarayanan, 2012 ^i^ [56]

						Subramanian et al., 2013 ^j^ [47]
						Saha et al., 2012 ^k^ [92]
						Mondal et al., 2019 ^l^ [15]
						Nirmalraj et al., 2015 ^m^ [93]
						Reddy et al., 2013 ^n^ [94]
						Reddy et al., 2015 ^o^ [16]
						Ayob; Saari; Samad, 2012 ^p^ [95]
						Indrayoni et al., 2016 ^q^ [96]
						Phatangare et al., 2017 ^r^ [97]
						Kumar et al., 2018 ^s^ [98]
						Bhagya; Chandrashekar, 2013 ^t^ [20]

						Kiren et al., 2014 ^u^ [99]
						Ningsih et al., 2015 ^v^ [100]
						Souza et al., 2017 ^x^ [48]
						Bhagya; Chandrashekar; Kalluraya, 2013 ^w^ [65]

						Sinansari; Prajogo; Widiyanti, 2018 ^y^ [27]

						Prasad, 2014 ^z^ [60]
						Sulistyowati et al., 2017 ^a′^ [101]
						Patel; Zaveri, 2014 ^b′^ [50]
						Widiyanti; Prajogo; Widodo, 2018 ^c′^ [30]

						Widiyanti; Prajogo; Hikmawanti, 2016 ^d′^ [66]

						Supparmaniam; Bohari, 2015 ^e′^ [102]

						Widodo; Widiyanti; Prajogo, 2018 ^f′^ [28]

						Prajogo; Widiyanti; Riza, 2016 ^g′^ [29]

						Prajogo et al., 2015 ^h′^ [26]
						Zhang et al., 2017 ^i′^ [25]
						Mnatsakanyan et al., 2018 ^j′^ [103]
						Varma et al., 2011 ^k′^ [104]
						Bhavana et al., 2020 ^l′^ [105]
						Ratih et al., 2019 ^m′^ [106]
						Zhang et al., 2020 ^n′^ [107]
						Ramya, 2020 ^o′^ [108]
*Justicia graciliflora* (Standndl.) D.N. Gibson	Aerial parts ^a^	NI ^a^	Yes ^a^	Yes ^a^	Panama ^a^	Calderón et al., 2012 ^a^ [57]

*Justicia hypocrateriformis* Vahl	Leaf ^a^	Aqueous ^a^	Yes ^a^	Yes ^a^	Camaeroon ^a^	Agbor et al., 2014 ^a^ [109]

*Justicia insularis* T. Anderson	Leaf ^a,b,c,d,e^	Aqueous ^a,b,c^	Yes ^b,c,d^	Yes ^a,b,c,d^	Cameroon ^a,b,c^	Telefo et al., 2012 ^a^ [110]
		Methanol ^d^	No ^a^		Nigeria ^d^	Mbemya et al., 2018 ^b^ [111]
						Goka et al., 2016 ^c^ [112]
						Fadayomi et al., 2021 ^d^ [113]
*Justicia neesii* Ramamoorthy	Whole plant ^a^	Ethanol ^a^	Yes ^a^	Yes ^a,b^	India ^a,b^	Sridhar; Duggirala; Puchchakayala, 2014 ^a^ [114]
			No^b^			
						Sridhar; Lakshmi; Goverdham, 2015 ^b^ [115]

*Justicia nodicaulis* (Nees) Leonard	Leaf ^a^	NI ^a^	Yes ^a^	No ^a^	Brazil ^a^	Rocha; Peixoto; Santos, 2019 ^a^ [61]


*Justicia paracambi* Braz	Leaf ^a^	Aqueous ^a^	No ^a^	Yes ^a^	Brazil ^a^	Azevedo Junior et al., 2022 ^a^ [116]
*Justicia pectoralis* Jacq.	Leaf ^a,d,e,f,g,k,n^	Hydroalcoholic ^b,c,h^	Yes ^a,b,c,f,g,h,i,j,k,l,m^	Yes ^a,b,c,d,e,f,h,i,j,k,l,n^	Brazil ^a,b,c,d,f,g,h,i,j,k,l,m,n^	Provensi, 2018 ^a^ [117]
	Aerial parts ^b,c,h,i,j,l,m^	Aqueous ^d,e,k,n^	No ^d,e,n^	No ^g,m^	India ^e^	Venâncio, 2015 ^b^ [118]
		Methanol ^f^				Silva, 2018 ^c^ [119]
		NI ^g^				Furtado et al., 2015 ^d^ [120]
		Ethanol ^i,j,l,m^				Cameron et al., 2015 ^e^ [121]
		Hydroketone ^k^				Cassola et al., 2019 ^f^ [19]
						Vargem, 2015 ^g^ [67]
						Moura et al., 2017 ^h^ [122]
						Lima, 2017 ^i^ [52]
						Rodrigues, 2017 ^j^ [123]
						Nunes et al., 2018 ^k^ [124]
						Carvalho et al., 2020 ^l^ [125]
						Lima et al., 2020 ^m^ [126]
						Guimarães et al., 2020 ^n^ [127]
*Justicia procubens* L.	Whole plant ^a,b,f,g,h,l,q,s^	Ethanol ^a,b,c,f,g,h,j,m,n,o,r,s^	Yes ^a,b,c,d,e,f,g,h,j,l,m,n,o,p,r,s^	Yes ^c,d,e,f,g,i,j,k,l,n,p,q,s^	China ^a,b,d,e,f,g,h,i,l,m,n,o,s^	Xiong et al., 2020 ^a^ [63]
	NI ^c,d,e,i,m,n,o,p,r^	NI ^d,i,p^	No ^i,k,q^	No ^a,b,h,m,o,r^	South Korea ^c,j,r^	Jiang et al., 2017 ^b^ [128]
	Aerial parts ^j^	Aqueous ^j^			India ^k^	Youm et al., 2018 ^c^ [129]
	Leaf ^k^	Methanol ^e,k,m,q^			Taiwan ^p,q^	Luo et al., 2014 ^d^ [130]
		Ethyl acetate ^l^				Luo et al., 2013 ^e^ [131]
						Jin et al. 2014 ^f^ [132]
						Jin et al.,2015 ^g^ [133]
						Jin; Yang; Dong, 2016 ^h^ [134]
						He et al., 2012 ^i^ [135]
						Youm et al., 2017 ^j^ [62]
						Kamaraj et al., 2012 ^k^ [136]
						Liu et al., 2018 ^l^ [137]
						Luo et al., 2013 ^m^ [131]
						Luo et al., 2016 ^n^ [138]
						Zhou et al., 2015 ^o^ [139]
						Wang et al., 2015 ^p^ [140]
						Won et al., 2014 ^q^ [141]
						Lee et al., 2020 ^r^ [142]
						Lv et al., 2020 ^s^ [143]
*Justicia refractifolia* (Kuntze) Leonard	Stem and leaf ^a^	NI ^a^	Yes ^a^	Yes ^a^	Panama ^a^	Calderón et al., 2012 ^a^ [57]

*Justicia schimperiana* T. Anderson	Leaf ^a,b,c,d^	Methanol ^a,b,c,d^	Yes ^a,b,c,d^	Yes ^a,b,c,d^	Ethiopia ^a,b,c,d^	Mekonnen; Asrie; Wubneh, 2018 ^a^ [144]

						Tesfaye, 2017 ^b^ [145]
						Abdela; Engidawork; Shibeshi, 2014 ^c^ [146]

						G/giorgis et al., 2022 ^d^ [147]
*Justicia secunda* Vahl	Leaf ^a,c,d,e,f,g,h,i,k,l,m,n,o,p^	Methanol ^a,d,e,f,j,k,n,p^	Yes ^b,c,e,f,g,i,j,k,l,m,n,o^	Yes ^a,b,c,d,e,f,g,h,k,n,p^	Nigeria ^a,d,f,g,k,l,m,n,o,p^	Onoja et al., 2017 ^a^ [148]
	Stem, Leaf and Root ^b^	NI^bmo^	No ^a,d,h,p^	No ^i,j,l,m,o^	Panama ^b^	Calderón et al., 2012 ^b^ [57]
	Aerial parts ^j^	Ethanol ^c,n^			Benin ^c^	Moukimoul et al., 2014 ^c^ [149]
		Ethyl acetate ^e^			Ghana ^e^	Anyasor; Okanlawon; Ogunbiyi, 2019 ^d^ [150]
		Aqueous ^e,g,h,i^			Ivory Coast ^h,i^	
		Hexane ^l^			Ecuador ^j^	Yamoah et al., 2020 ^e^ [53]
						Osioma; Hamilton-Amachree, 2017 ^f^ [151]

						Anyasor; Moses; Kale, 2020 ^g^ [152]
						Abo; Kouakou; Yapo, 2016 ^h^ [153]
						Koffi et al., 2013 ^i^ [154]
						Theiler et al., 2014 ^j^ [155]
						Aimofumeh; Anyasor; Esiaba, 2020 ^k^ [156]

						Ajuru et al., 2022 ^l^ [84]
						Arogbodo, 2020 ^m^ [157]
						Ayodele; Odusole; Adekanmbi, 2020 ^n^ [158]
						Odokwo; Onifade, 2020 ^o^ [159]
						Ofeimun; Enwerem; Benjamin, 2020 ^p^ [160]
*Justicia simplex* D. Don.	Aerial parts ^a^	Ethanol ^a,c^	Yes ^a,c^	Yes ^a,b,c^	India ^a,b,c^	Joseph et al., 2017 ^a^ [161]
	Whole plant ^b^	Petroleum ether ^a^	No ^b^			Kumaran et al., 2013 ^b^ [162]
	Leaf ^c^	Methanol ^b^				Eswari et al., 2014 ^c^ [163]
		Benzene ^c^				
		Aqueous ^c^				
		Hexane ^c^				
*Justicia spicigera* Schltdl	Leaf ^a,b,d,f,h,i,j,l,n,p,q^	Ethanol ^a,b,c,d,f,g,h,j,k^	Yes ^a,b,c,d,e,f,g,h,i,j,m,n,o,p,q,r^	Yes ^a,b,c,d,e,f,g,h,i,j,k,l,m,n,o,pq,r^	Mexico ^a,b,c,d,e,f,h,i,j,k,l,m,o,p,q,r^	Ángeles-López et al., 2019 ^a^ [164]
	Whole plant ^c^	Chloroform ^e^	No ^k,l^		Egypt ^g^	Cassani et al., 2014 ^b^ [165]
	Aerial parts ^e,f,m,o,r^	Aqueous ^i,j,m^			Ecuador ^n^	Vega-Avila et al., 2012 ^c^ [166]
	NI ^k^	Methanol ^l,n^				Ortiz-Andrade et al., 2012 ^d^ [42]
		Hydroalcoholic ^o,p,q^				Esquivel-Gutiérrez et al., 2013 ^e^ [64]
		Ethyl acetate ^r^				
						Zapata-Morales et al., 2016 ^f^ [167]
						Awad et al., 2015 ^g^ [168]
						Alonso-Castro et al., 2012 ^h^ [22]
						García-Ríos et al., 2019 ^i^ [169]
						Baqueiro-Peña; Gerrero-Beltrán, 2017 ^j^ [170]

						Israel et al., 2017 ^k^ [171]
						Magos-Guerrero; Santiago-Mejía; Carrasco, 2017 ^l^ [172]

						González-Trujano et al. 2017 ^m^ [173]

						Theiler et al., 2016 ^n^ [174]
						Fernández-Pomares et al., 2018 ^o^ [175]

						Hernández-Rodríguez et al., 2020 ^p^ [176]
						Castro-Alatorre et al., 2021 ^q^ [177]
						Pérez-Vásquez et al., 2022 ^r^ [178]
*Justicia subsessilis* Oliv.	Aerial parts ^a^	Hexane ^a^	Yes ^a^	Yes ^a^	Burundi ^a^	Ngezahayo et al. 2017 ^a^ [179]
		Dichloromethane ^a^				
		Ethyl acetate ^a^				
		Methanol ^a^				
		Aqueous ^a^				
*Justicia thunbergioides* (Lindau) Leonard	Leaf ^a,b^	Hexane ^a^	Yes ^a,b^	Yes ^a,b^	Brazil ^a,b^	Provensi, 2018 ^a^ [117]
		Dichloromethane^a^				Vasconcelos, 2019 ^b^ [39]
		Methanol ^a^				
		Hydroalcoholic ^b^				
*Justicia tranquebariensis* L.	Aerial parts ^a,b^	Ethanol ^ac^	Yes ^a,c,d^	Yes ^a,b,c,d,e^	India ^a,b,c,d^	Senthamari; Akilandeswari; Valarmathi, 2013 ^a^ [55]
	NI ^c^	Aqueous ^abde^	No ^b,e^		Malaysia ^e^	
	Leaf ^d,e^	Hexane ^c^				Radhika et al., 2013 ^b^ [180]
						Krishnamoorthi; Ratha Bai, 2015 ^c^ [181]

						Krishnamoorthi, 2015 ^d^ [182]
						Sukalingam; Ganesan; Xu, 2018 ^e^ [183]

*Justicia vahlii* Roth	Whole plant ^a,b^	Buthanolic ^a^	Yes ^a,b^	Yes ^a,b^	Pakistan ^a,b^	Basit et al., 2022 ^a^ [184]
		Hydroalcoholic ^b^				Basit et al., 2022 ^b^ [185]
*Justicia wasshauseniana* Profice	Aerial parts ^a^	Methanol ^a^	Yes ^a^	Yes ^ab^	Brazil ^a,b^	Fernandes, 2016 ^a^ [186]
	Leaf ^b^	Dichlorometane ^a^	No ^b^			Azevedo Junior et al., 2022 ^b^ [116]
		Hydroalcoholic ^b^				
		Aqueous ^b^				
*Justicia wynaandensis* B. Heyne	Leaf ^a,b,c^	Methanol ^a,b,c^			India ^a,b,c^	Dsouza; Nanjaiah, 2018 ^a^ [187]
		Ethyl acetate ^b^	Yes ^a,b^	Yes ^a,c^		Ponnamma; Manjunath, 2012 ^b^ [51]
		Dichloromethane ^c^	No ^c^	No ^b^		
						Zameer et al., 2016 ^c^ [188]

NI—Not Informed; The letters that are in superscript refer to the authors of the references.

**Table 2 molecules-28-01190-t002:** Biological activities of isolated secondary metabolites from species of *Justicia*.

Compound	Biological Activities	Species	Tested Parts	Extract	Reference
Glycosylated β-sitosterol (1)	Anti-inflammatory	*J. acuminatissima* (Miq.) Bremek.	Aerial parts	Ethanol	Corrêa et al., 2014 [34]
Glycosylated stigmasterol (2)	Anti-inflammatory	*J. acuminatissima* (Miq.) Bremek.	Aerial parts	Ethanol	Corrêa et al., 2014 [34]
Phytol (3)	Anti-inflammatory	J. gendarussa Burm.	Leaves	NI	Phantagare et al., 2017 [189]
Apigenin (4)	Anti-inflammatory	*J. gendarussa* Burm.	Root	Methanol	Kumar et al., 2018 [98]
Naringenin (5)	Cytotoxic	*J. gendarussa* Burm.	Leaves	Methanol	Ayob; Samad; Bohari, 2013 [24]
Kaempferol (6)	Cytotoxic	*J. gendarussa* Burm.	Leaves	Methanol	Ayob; Samad; Bohari, 2013 [24]
3,3′,4′-Trihydroxyflavone (7)	Antimicrobial	*J. wynaadensis* B. Heyne	Leaves	Methanol	Dsouza; Nanjaiah, 2018 [187]
Vasicoline (8)	Antimicrobial	*J. adhatoda* L.	Leaves	Methanol	Jha et al., 2012 [37]
Vasicine (9)	Antimicrobial, antioxidant and anticancerous	*J. adhatoda* L.	Leaves	Methanol and Hydroalcoholic	Pa; Mathew, 2012 [36]; Kaur et al., 2016 [9]

Etamine (10)	Antioxidant	*J. gendarussa* Burm.	Aerial parts	Ethanol	Zhang et al., 2020 [109]
Secundallerone B (11)	Antidiabetic	*J. secunda* Vahl.	Leaves	Methanol	Theiler et al., 2016 [174]
Secundallerone C (12)	Antidiabetic	*J. secunda* Vahl.	Leaves	Methanol	Theiler et al., 2016 [174]
2-caffeoyloxy-4-hydroxy-glutaric acid (13)	Antidiabetic	*J. secunda* Vahl.	Leaves	Methanol	Theiler et al., 2016 [174]

Kaempferitrin (14)	Antinociceptive, cytotoxic, antidiabetic and anticonvulsant	*J. spicigera* Schltdl.	Aerial parts; Leaves	Ethanol and aqueous	Cassani et al., 2014 [165]; Ángeles-López et al., 2019 [164]; Zapata-Morales et al., 2016 [167]; Alonso-Castro et al., 2012 [22]; Ortiz-Andrade et al., 2012 [42]; González-Trujano et al., 2017 [173]

Gendarussin A (15)	Cytotoxic	*J. gendarussa* Burm.	Leaves	Ethanol	Prajogo et al., 2015 [26]
6′-hydroxyl justicidin A (16)	Cytotoxic	*J. procumbens* L.	Whole plant	Ethanol	Jin et al., 2014 [132]
6′-hydroxyl justicidin B (17)	Cytotoxic	*J. procumbens* L.	Whole plant	Ethanol	Jin et al., 2014 [132]
6′-hydroxyl justicidin C (18)	Cytotoxic	*J. procumbens* L	NI	Ethanol	Luo; Kong; Yang, 2014 [190]
Justicidin A (19)	Cytotoxic, pharmacokinetics, anti-inflammatory and anti-allergic	*J. procumbens* L.	Aerial parts; NI	Methanol and ethanol	Won et al., 2014 [141]; Youm et al., 2017 [62]; Youm et al., 2018 [129]; Wang et al., 2015 [140]

Chinensinaphthol methyl ether (20)	Cytotoxic	*J. procumbens* L.	NI	Ethanol	Luo et al., 2014 [130]
Taiwanin E methyl ether (21)	Cytotoxic	*J. procumbens* L.	NI	Ethanol	Luo et al., 2014 [130]
Paclitaxel (22)	Cytotoxic	*J. procumbens* L.	NI	Ethanol	Luo et al., 2014 [130]
Podophyllotoxin (23)	Cytotoxic	*J. procumbens* L.	NI	Ethanol	Luo et al., 2014 [130]
Justicidin B (24)	Pharmacokinetics, anti-inflammatory and anti-allergic	*J. procumbens* L.	NI; Aerial parts	Ethanol	Luo et al., 2014 [130]; Luo et al., 2016 [138]; Youm et al., 2017 [62]; Youm et al., 2018 [129]

Justicidin C (25)	Anti-inflammatory and cytotoxic	*J. procumbens* L.	NI; Aerial parts	Ethanol	Youm et al., 2017 [62]; Luo; Kong; Yang, 2014 [190]

Phyllamyricin C (26)	Anti-inflammatory	*J. procumbens* L.	Aerial parts	Ethanol	Youm et al., 2017 [61]
Pronaphthalide A (27)	Cytotoxic	*J. procumbens* L.	Whole plant	Ethanol	Jin et al., 2014 [132]
Procumbenoside J (28)	Cytotoxic	*J. procumbens* L.	Whole plant	Ethanol	Jin et al., 2014 [132]
Tuberculatin (29)	Cytotoxic	*J. procumbens* L.	Whole plant	Ethanol	Jin et al., 2014 [132]
Diphyllin (30)	Cytotoxic	*J. procumbens* L.	Whole plant	Ethanol	Jin et al., 2014 [132]; Lv et al., 2020 [143]
Procumbenoside H (31)	Cytotoxic	*J. procumbens* L.	Whole plant	Ethanol	Jin et al., 2015 [133]
(+)-pinoresinol (32)	Antioxidant	*J. gendarussa* Burm.	Aerial parts	Ethanol	Zhang et al., 2020 [107]
2′-methoxy-4″-hydroxydimetoxykobusin (33)	Anti-inflammatory	*J. gendarussa* Burm.	Aerial parts	Ethanol	Zhang et al., 2020 [107]
Brazoide A (34)	Anti-inflammatory	*J. gendarussa* Burm.	Aerial parts	Ethanol	Zhang et al., 2020 [107]
Justiprocumin A (35)	Cytotoxic	*J. gendarussa* Burm.	Stem	Methanol	Zhang et al., 2017 [25]
Justiprocumin B (36)	Cytotoxic	*J. gendarussa* Burm.	Stem	Methanol	Zhang et al., 2017 [25]
Patentiflorin A (37)	Cytotoxic	*J. gendarussa* Burm.	Stem and root	Methanol	Zhang et al., 2017 [25]
Triacontanoic ester of 5-hydroxyjustisolin (38)	Cytotoxic	*J. simplex* D.Don.	Aerial parts	Petroleum ether	Joseph et al., 2017 [161]

16(α/β)-hydroxy-cleroda-3,13 (14)Z-dien-15,16-olide (39)	Cytotoxic	*J. insularis* T. Anderson	Leaves	Methanol	Fadayomi et al., 2021 [113]

16-oxo-cleroda-3,13(14)E-dien-15-oic acid (40)	Cytotoxic	*J. insularis* T. Anderson	Leaves	Methanol	Fadayomi et al., 2021 [113]

Justicianene D (41)	Cytotoxic	*J. procumbens* L.	Whole plant	Ethanol	Lv et al., 2020 [143]
2-N-(p-coumaroyl)-3*H*-phenoxazin-3-one (42)	Enzyme inhibitor	*J. spicigera* Schltdl.	Aerial parts	Ethyl acetate	Pérez-Vásquez et al., 2022 [178]
3″-*O*-acetyl-kaempferitrin (43)	Enzyme inhibitor	*J. spicigera* Schltdl.	Aerial parts	Ethyl acetate	Pérez-Vásquez et al., 2022 [178]
kaempferol 7-O-α-L-rhamnopyranoside (44)	Enzyme inhibitor	*J. spicigera* Schltdl.	Aerial parts	Ethyl acetate	Pérez-Vásquez et al., 2022 [178]
perisbivalvine B (45)	Enzyme inhibitor	*J. spicigera* Schltdl.	Aerial parts	Ethyl acetate	Pérez-Vásquez et al., 2022 [178]
2,5-dimethoxy-p-benzoquinone (46)	Enzyme inhibitor	*J. spicigera* Schltdl.	Aerial parts	Ethyl acetate	Pérez-Vásquez et al., 2022 [178]

NI—Not Informed.

**Table 3 molecules-28-01190-t003:** Experimental details and refinement data of naringenin.

Crystal Data	Naringenin
Chemical formula	C_15_H_12_O_5_
Formula weight	272.25
Crystal system, space group	Monoclinic, P2_1_/c
a, b, c (Å)	4.965 (3) 15.449 (6) 16.845 (8)
α = β = γ (°)	90.00 103.86(8) 90.00
V (Å^3^)	1254.5(12)
Z	4
ρ_calc_ g/cm^3^	1.441
µ (mm^−1^)	0.109
F (000)	568.0
Radiation type	MoKα (λ = 0.71073)
Final R indexes [I ≥ 2σ (I)]	R_1_ = 0.0540, wR_2_ = 0.0540

**Table 4 molecules-28-01190-t004:** Relevant experimental bond length (Å), bond angles (°), and dihedral angles (°) for naringenin.

Naringenin			
O1-C1	1.45	O2-C3-C2	119.73
O1-C9	1.37	C9-O1-C1-C2	49.81
O2-C3	1.25	C1-O1-C9-C8	156.56
O3-C5	1.35	C1-O1-C9-C4	−24.81
O4-C7	1.36	O1-C1-C10-C11	120.51
O5-C13	1.38	C2-C1-C10-C11	−115.58
C1-C2	1.51	C2-C1-C10-C15	62.47
C1-C10	1.51	O1-C1-C10-C15	−61.44
C1-C10-C11	119.71	C1-C2-C3-C4	−50.62
O1-C1-C10	108.25	C1-C2-C3-O2	−153.63

**Table 5 molecules-28-01190-t005:** Hydrogen-bond geometry (Å, º) for naringenin.

	*D*–H···*A*	*D*–H	H···*A*	*D*···*A*	*D*–H···*A*	Symmetry Code
Naringenin	O3–H3···O2	0.86	1.88	2.648	147	INTRA
	O4–H4···O5	0.83	2.04	2.805	154	1 − x, −1 − y, 1 − z
	O5–H5···O2	0.77	1.95	2.711	172	1 + x, 1/2 − y, 1/2 + z
	C15–H15···O4	0.88	2.59	3.417	155	1 − x, −1/2 + y, 1/2 − z

## Data Availability

Not applicable.

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
