# Peer review of "Overview of the Justicia Genus: Insights into Its Chemical Diversity and Biological Potential"

_molecules, 2023, doi:10.3390/molecules28031190_

Round 1

Reviewer 1 Report

This is an interesting overview of the collection of natural products derived from Judiscia with potential biological and medical activities. The paper gives an excellent background of the source of these plants from which the compounds are derived and the compounds were identified from the literature citation.However, the computational analysis of the compounds at current stage is limited and should be further expanded.

1.The authors provide some computational analysis of selected compounds like Nargingenin but in the crystal/solid states. This has minimal biological significance as compound exhibit much greater conformational diversity in their physiological condition to enable target binding. Furthermore, biological active conformation could be very different from their crystalline state. In addition to the crystalline state,  I would suggest some conformational analysis to identify/predict the physiological conformation of selected compounds.   

2. For the compound structures listed in figure 2, could the authors group the compounds based on their types and mechanisms and perhaps provide some annotations in a table?

3. Along the above points, I would recommend additional diversity analysis of compound based on the structural fingerprints of the genus. Perhaps can be further visualized using dimensional reduction technique such as PCA, MDS or tSNE to see how many groups or type of mechanisms are there.

4. On the other hand, the sub title of section 3.3 is a bit misleading as the section is about the chemical reactivity of Nargingenin analyzed by molecular orbital theory but nothing has been said about the chemical diversity of the natural products.

Minor point:

-species, compounds, and their effects

Reviewer 2 Report

Following heading may assist author to improve their review.

1. The word already mentioned in the title should not be repeated in keywords. Better to select the correct keywords related to the study.

2. Author should mandatorily incorporate method of review, nothing is mentioned about the literature extraction, selection and appropriate citations. Prisma format is generally preffered for review. The author can go through it, Preferred Reporting Items for Systematic Reviews and Meta-Analysis (PRISMA).

3. Toxicity or protective activity of all the reported compounds needs to be mentioned under the discussion part to have a clear picture of toxicological profiles.

4. Reported LD50 & IC50 should be incorporated, if available.
